# Reinforcement Learning with Competitive Ensembles of Information-Constrained Primitives

**Anirudh Goyal[1], Shagun Sodhani[2], Jonathan Binas[1], Xue Bin Peng[3],**
**Sergey Levine[3], Yoshua Bengio[1]**

## Abstract

Reinforcement learning agents that operate in diverse and complex environments can benefit from the structured decomposition of their behavior. Often, this is addressed in the context of hierarchical reinforcement learning, where the aim is to decompose a policy into lower-level primitives or *options*, and a higher-level meta-policy that triggers the appropriate behaviors for a given situation. However, the meta-policy must still produce appropriate decisions in all the states. In this work, we propose a policy design that decomposes into primitives, similarly to hierarchical reinforcement learning, but without an explicit high-level meta-policy. Instead, each primitive can decide for themselves whether they wish to act in the current state. We use an information-theoretic mechanism for enabling this decentralized decision: each primitive chooses how much information it needs about the current state to make a decision and the primitive that requests the most information about the current state acts in the environment. Regularizing the primitives to use as little information as possible leads to natural competition and specialization. We experimentally demonstrate that this policy architecture improves over both flat and hierarchical policies in terms of generalization.

## 1 Introduction

Learning policies that generalize to new environments or tasks is a fundamental challenge in reinforcement learning. While deep reinforcement learning has enabled training powerful policies, which outperform humans on specific, well-defined tasks (Mnih et al., 2015), their performance often diminishes when the properties of the environment or the task change to regimes not encountered during training.

This is in stark contrast to how humans learn, plan, and act: humans can seamlessly switch between different aspects of a task, transfer knowledge to new tasks from remotely related but essentially distinct prior experience, and combine primitives (or skills) used for distinct aspects of different tasks in meaningful ways to solve new problems. A hypothesis hinting at the reasons for this discrepancy is that the world is inherently compositional, such that its features can be described by compositions of small sets of primitive mechanisms (Parascandolo et al., 2017). Since humans seem to benefit from learning skills and learning to combine skills, it might be a useful inductive bias for the learning models as well.

This is addressed to some extent by the hierarchical reinforcement learning (HRL) methods, which focus on learning representations at multiple spatial and temporal scales, thus enabling better exploration strategies and improved generalization performance (Dayan & Hinton, 1993; Sutton et al., 1999b; Dietterich, 2000; Kulkarni et al., 2016). However, hierarchical approaches rely on some form of learned high-level controller, which decides when to activate different components in the hierarchy. While low-level sub-policies can specialize to smaller portions of the state space, the top-level controller (or master policy) needs to know how to deal with any given state. That is, it should provide optimal behavior for the entire accessible state space. As the master policy is trained

---

[1] Mila, University of Montreal; [2] Facebook AI Research; work done while the author was at Mila, University of Montreal; [3] University of California, Berkeley. `anirudhgoyal9119@gmail.com`

on a particular state distribution, learning it in a way that generalizes to new environments effectively becomes the bottleneck for such approaches (Sasha Vezhnevets et al., 2017; Andreas et al., 2017).

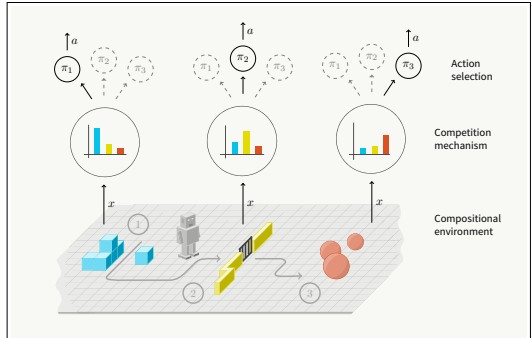 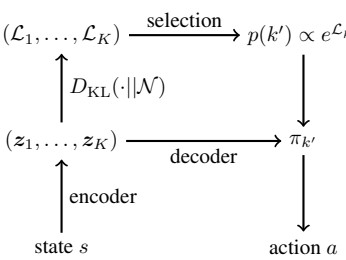

Figure 1: Illustration of our model (Left): An intrinsic competition mechanism, based on the amount of information each primitive requests, is used to select a primitive to be active for a given input. Each primitive focuses on distinct features of the environment; in this case, one policy focuses on boxes, a second one on gates, and the third one on spheres. Right: The primitive-selection mechanism of our model. The primitive with most information acts in the environment and gets the reward.

We argue, and empirically show, that in order to achieve better generalization, the interaction between the low-level primitives and the selection thereof should itself be performed without requiring a single centralized network that understands the entire state space. We, therefore, propose a decentralized approach as an alternative to standard HRL, where we only learn a set of low-level primitives without learning an explicit high-level controller. In particular, we construct a factorized representation of the policy by learning simple *primitive* policies, which focus on distinct regions of the state space. Rather than being gated by a single meta-policy, the primitives directly compete with one another to determine which one should be active at any given time, based on the degree to which their state encoders "recognize" the current state input. While, technically, the competition between primitives implicitly realizes a global selection mechanism, we consider our model *decentralized* in the sense that individual primitives can function on their own, and can be combined in new ways, without relying on an explicit high-level controller.

We frame the problem as one of information transfer between the current state and a dynamically selected primitive policy. Each policy can, by itself, decide to request information about the current state, and the amount of information requested is used to determine which primitive acts in the current state. Since the amount of state information that a single primitive can access is limited, each primitive is encouraged to use its resources wisely. Constraining the amount of accessible information in this way naturally leads to a decentralized competition and decision mechanism where individual primitives specialize in smaller regions of the state space. We formalize this information-driven objective based on the variational information bottleneck. The resulting set of competing primitives achieves both a meaningful factorization of the policy and an effective decision mechanism for which primitives to use. Importantly, not relying on a centralized meta-policy enables the individual primitive mechanisms can be recombined in a *plug-and-play* fashion, and the primitives can be transferred seamlessly to new environments.

**Contributions:** In summary, the contributions of our work are as follows: (1) We propose a method for learning and operating a set of functional primitives in a decentralized way, without requiring an explicit high-level meta-controller to select the active primitives (see Fig. 1 for illustration). (2) We introduce an information-theoretic objective, the effects of which are twofold: a) it leads to the specialization of individual primitives to distinct regions of the state space, and b) it enables a competition mechanism, which is used to select active primitives in a decentralized manner. (3) We demonstrate the superior transfer learning performance of our model, which is due to the flexibility of the proposed framework regarding the dynamic addition, removal, and recombination of primitives. Decentralized primitives can be successfully transferred to larger or previously unseen environments, and outperform models with an explicit meta-controller for primitive selection.

## 2 PRELIMINARIES

We consider a Markov decision process (MDP) defined by the tuple $(\mathcal{S}, \mathcal{A}, P, r, \gamma)$, where the state space $\mathcal{S}$ and the action space $\mathcal{A}$ may be discrete or continuous. The environment emits a bounded reward $r : \mathcal{S} \times \mathcal{A} \rightarrow [r_{min}, r_{max}]$ on each transition and $\gamma \in [0, 1)$ is the discount factor. $\pi(.|s)$ denotes a policy over the actions given the current state $s$. $R(\pi) = \mathbb{E}_\pi[\sum_t \gamma^t r(s_t)]$ denotes the expected total return when an agent follows the policy $\pi$. The standard objective in reinforcement learning is to maximize the expected total return $R(\pi)$. We use the concept of the information bottleneck (Tishby et al., 2000) to learn compressed representations. The information bottleneck objective is formalized as minimizing the mutual information of a *bottleneck* representation layer with the input while maximizing its mutual information with the corresponding output. This type of input compression has been shown to improve generalization (Achille & Soatto, 2016; Alemi et al., 2016).

## 3 INFORMATION-THEORETIC LEARNING OF DISTINCT PRIMITIVES

Our goal is to learn a policy, composed of multiple primitive sub-policies, to maximize the expected reward over $T$-step interactions for a distribution of tasks. Simple primitives which focus on solving a part of the given task (and not the complete task) should generalize more effectively, as they can be applied to similar aspects of different tasks (subtasks) even if the overall objective of the tasks are drastically different. Learning primitives in this way can also be viewed as learning a factorized representation of a policy, which is composed of several *independent* policies.

Our proposed approach consists of three mechanisms: 1) a mechanism for restricting a particular primitive to a subset of the state space; 2) a competition mechanism between primitives to select the most effective primitive for a given state; 3) a regularization mechanism to improve the generalization performance of the policy as a whole. We consider experiments with both fixed and variable sets of primitives and show that our method allows for primitives to be added or removed during training, or recombined in new ways. Each primitive is represented by a differentiable, parameterized function approximator, such as a neural network.

### 3.1 PRIMITIVES WITH AN INFORMATION BOTTLENECK

To encourage each primitive to encode information from a particular part of state space, we limit the amount of information each primitive can access from the state. In particular, each primitive has an information bottleneck with respect to the input state, preventing it from using all the information from the state.

We define the overall policy as a mixture of primitives,

$$\pi(a \mid s) = \sum_k c_k \pi^k(a \mid s),$$

where $\pi^k(a \mid s)$ denotes the $k^{\text{th}}$ primitive and $c_k = \delta_{kk'}$ for $k' \sim p(k' \mid s)$. We denote the probability of selecting the $k^{\text{th}}$ primitive as $\alpha_k(s) := p(k \mid s)$.

Rather than learning an explicit model for $p(k \mid s)$, however, we impose an information-based mechanism for selecting primitives, wherein we limit the amount of information each primitive can contain and select the ones that request the most information about the state. To implement an information bottleneck, we design each of the $K$ primitives to be composed of an encoder $p_{\text{enc}}(z_k \mid s)$ and a decoder $p_{\text{dec}}(a \mid z_k)$, together forming the primitive policy,

$$\pi_\theta^k(a \mid s) = \int_z p_{\text{enc}}(z_k \mid s) \, p_{\text{dec}}(a \mid z_k) \, \mathrm{d}z_k .$$

The encoder output $z_k$ is meant to represent the information about the current state $s$ that an individual primitive $k$ believes is important to access in order to perform well. The decoder takes this encoded information and produces a distribution over the actions $a$. Following the variational information bottleneck objective (Alemi et al., 2016), we penalize the KL divergence of $p_{enc}(z_k|s)$ and a prior $p(z)$,

$$\mathcal{L}_k = \mathrm{D}_{\text{KL}} \left( p_{\text{enc}}(z_k \mid s) || p(z) \right) . \tag{1}$$

---

In practice, we estimate the marginalization over $z$ using a single sample throughout our experiments.

In other words, a primitive pays an "information cost" proportional to $\mathcal{L}_k$ for accessing the information about the current state.

In the experiments below, we fix the prior to be a unit Gaussian. In the general case, we can learn the prior as well and include its parameters in $\theta$. The information bottleneck encourages each primitive to limit its knowledge about the current state, but it will not prevent multiple primitives from specializing to similar parts of the state space. To mitigate this redundancy, and to make individual primitives focus on different regions of the state space, we introduce an information-based competition mechanism to encourage diversity among the primitives.

## 3.2 COMPETING INFORMATION-CONSTRAINED PRIMITIVES

We can use the information measure from equation 1 to define a selection mechanism for the primitives without having to learn a centralized meta-policy. The intuition is that the information content of an individual primitive encodes its effectiveness in a given state $s$ such that the primitive with the highest value $\mathcal{L}_k$ should be activated in that particular state.

In particular, we set $\alpha_k = Z^{-1} \exp(\beta \mathcal{L}_k)$ to obtain a distribution over $k$ as a function of the information content, activating the primitives with the highest information content. Here, $Z = \sum_k \exp(\beta \mathcal{L}_k)$ is a normalization constant. This mechanism enables competition between primitives, leading them to focus on parts of the state space that they "understand" well and letting others act in other parts.

**Trading reward and information.** To perform proper credit assignment, the environment reward is distributed to primitives according to their participation in the global decision, i.e. the reward $r_k$ given to the $k^{th}$ primitive is weighted by its selection coefficient, such that $r_k = \alpha_k r$, with $r = \sum_k r_k$. Hence, a primitive can potentially get a higher reward when deciding to act, but it also pays a higher price for accessing more information about the current state. The information bottleneck and the competition mechanism, when combined with the overall reward maximization objective, will lead to specialization of individual primitives to distinct regions in the state space. That is, each primitive should specialize in a part of the state space that it can reliably associate rewards with. Since the entire ensemble still needs to understand all of the state space for the given task, different primitives need to encode and focus on different parts of the state space.

## 3.3 REGULARIZING PRIMITIVE SELECTION

The objective described above will optimize the expected return while minimizing the information content of individual primitives. This is not sufficient, however, as it might lead to highly unbalanced outcomes: some primitives might be more active initially and learn to become even more active, completely disabling other primitives.

Thus, in addition to minimizing each primitive's absolute information content, we need to normalize their activity with respect to each other. To do so, we penalize their information content in proportion to their activation by adding a regularization term of the form

$$\mathcal{L}_{\text{reg}} = \sum_k \alpha_k \mathcal{L}_k \,. \tag{2}$$

Note that this can be rewritten (see Appendix A) as $\mathcal{L}_{\text{reg}} = -H(\alpha) + \text{LSE}(\mathcal{L}_1, \ldots, \mathcal{L}_K)$, where $H(\alpha)$ is the entropy of $\alpha$, and LSE is the *LogSumExp* function, $\text{LSE}(x) = \log(\sum_j e^{x_j})$. Thus, minimizing $\mathcal{L}_{\text{reg}}$ increases the entropy of $\alpha$, leading to a diverse set of primitive selections, in turn, ensuring that different combinations of the primitives are used. Similarly, LSE approximates the maximum of its arguments, $\text{LSE}(x) \approx \max_j x_j$, and, therefore, penalizes the dominating $\mathcal{L}_k$ terms, thus equalizing their magnitudes.

## 3.4 OBJECTIVE AND ALGORITHM SUMMARY

Our overall objective function consists of 3 terms,

1. The expected return from the standard RL objective, $R(\pi)$ which is distributed to the primitives according to their participation,

2. The individual bottleneck terms leading the individual primitives to focus on specific parts of the state space, $\mathcal{L}_k$ for $k = 1, \ldots, K$,

3. The regularization term applied to the combined model, $\mathcal{L}_{\text{reg}}$.

The overall objective for the $k^{th}$ primitive thus takes the form:

$$J_k(\theta) \equiv \mathbb{E}_{\pi_\theta}[r_k] - \beta_{\text{ind}}\mathcal{L}_k - \beta_{\text{reg}}\mathcal{L}_{\text{reg}}, \qquad (3)$$

where $\mathbb{E}_{\pi_\theta}$ denotes an expectation over the state trajectories generated by the agent's policy, $r_k = \alpha_k r$ is the reward given to the $k$th primitive, and $\beta_{\text{ind}}$, $\beta_{\text{reg}}$ are the parameters controlling the impact of the respective terms.

**Implementation:** In our experiments, the encoders $p_{\text{enc}}(z_k \mid s)$ and decoders $p_{\text{dec}}(a \mid z_k)$ (see. Fig. 1) are represented by neural networks, the parameters of which we denote by $\theta$. Actions are sampled through each primitive every step. While our approach is compatible with any RL method, we maximize $J(\theta)$ computed on-policy from the sampled trajectories using a score function estimator (Williams, 1992; Sutton et al., 1999a) specifically A2C (Mnih et al., 2016) (unless otherwise noted). Every experimental result reported has been averaged over 5 random seeds. Our model introduces 2 extra hyper-parameters $\beta_{\text{ind}}$, $\beta_{\text{reg}}$.

## 4 RELATED WORK

There are a wide variety of hierarchical reinforcement learning approaches(Sutton et al., 1998; Dayan & Hinton, 1993; Dietterich, 2000). One of the most widely applied HRL framework is the *Options* framework ((Sutton et al., 1999b)). An option can be thought of as an action that extends over multiple timesteps, thus providing the notion of temporal abstraction or subroutines in an MDP. Each option has its own policy (which is followed if the option is selected) and the termination condition (to stop the execution of that option). Many strategies are proposed for discovering options using task-specific hierarchies, such as pre-defined sub-goals (Heess et al., 2017), hand-designed features (Florensa et al., 2017), or diversity-promoting priors (Daniel et al., 2012; Eysenbach et al., 2018). These approaches do not generalize well to new tasks. Bacon et al. (2017) proposed an approach to learn options in an end-to-end manner by parameterizing the intra-option policy as well as the policy and termination condition for all the options. Eigen-options (Machado et al., 2017) use the eigenvalues of the Laplacian (for the transition graph induced by the MDP) to derive an intrinsic reward for discovering options as well as learning an intra-option policy.

In this work, we consider a sparse reward setup with high dimensional action spaces. In such a scenario, performing unsupervised pretraining or using auxiliary rewards leads to much better performance (Frans et al., 2017; Florensa et al., 2017; Heess et al., 2017). Auxiliary tasks such as motion imitation have been applied to learn motor primitives that are capable of performing a variety of sophisticated skills (Liu & Hodgins, 2017; Peng et al., 2017; Merel et al., 2019b;a). Our work is also related to the *Neural Module Network* family of architectures (Andreas et al., 2017; Johnson et al., 2017; Rosenbaum et al., 2019) where the idea is to learn *modules* that can perform some useful computation like solving a subtask and a *controller* that can learn to combine these modules for solving novel tasks. More recently, Wu et al. (2019) proposed a framework for using diverse suboptimal world models to learn primitive policies. The key difference between our approach and all the works mentioned above is that we learn functional primitives without requiring any explicit high-level meta-controller or master policy.

## 5 EXPERIMENTAL RESULTS

In this section, we briefly outline the tasks that we used to evaluate our proposed method and direct the reader to the appendix for the complete details of each task along with the hyperparameters used for the model. We designed experiments to address the following questions: **a) Learning primitives** – Can an ensemble of primitives be learned over a distribution of tasks? **b) Transfer Learning using primitives** – Can the learned primitives be transferred to unseen/unsolvable sparse environments? **c) Comparison to centralized methods** – How does our method compare to approaches where the primitives are trained using an explicit meta-controller, in a centralized way?

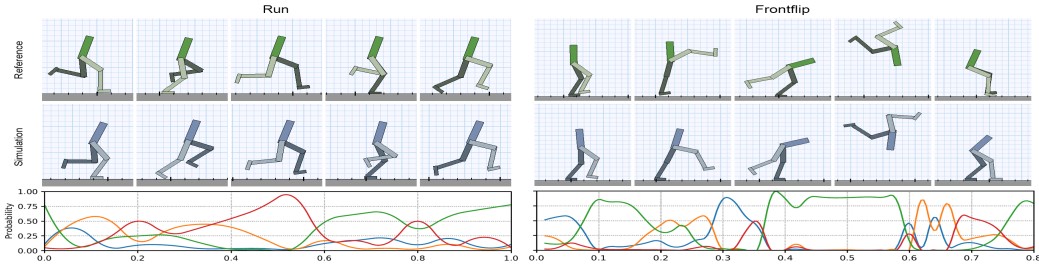

Figure 2: Snapshots of motions learned by the policy. **Top:** Reference motion clip. **Middle:** Simulated character imitating the reference motion. **Bottom:** Probability of selecting each primitive.

**Baselines.** We compare our proposed method to the following baselines: **a) Option Critic** (Bacon et al., 2017) – We extended the author's implementation of the Option Critic architecture and experimented with multiple variations in terms of hyperparameters and state/goal encoding. None of these yielded reasonable performance in partially observed tasks, so we omit it from the results. **b) MLSH** (Meta-Learning Shared Hierarchy) (Frans et al., 2017) – This method uses meta-learning to learn sub-policies that are shared across tasks along with learning a task-specific high-level master. It also requires a phase-wise training schedule between the master and the sub-policies to stabilize training. We use the MLSH implementation provided by the authors. **c) Transfer A2C:** In this method, we first learn a single policy on the one task and then transfer the policy to another task, followed by fine-tuning in the second task.

### 5.1 LEARNING ENSEMBLES OF FUNCTIONAL PRIMITIVES

We evaluate our approach on a number of RL environments to demonstrate that we can indeed learn sets of primitive policies focusing on different aspects of a task and collectively solving it.

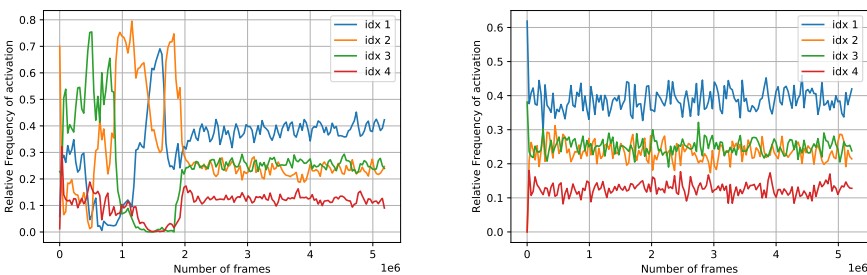

Figure 3: **Convergence of four primitives on Four Room Maze:** Left: We trained four primitives on the Four Room Maze task, where the goal was sampled from one of the two fixed goals. We see that the proposed algorithm is able to learn four primitives. Right: We transfer the learned primitives to the scenario where the goal is sampled from one of the four possible goals. The checkpointed model is ran on 100 different episodes (after a fixed number of steps/updates) and the normalized frequency of activation of the different primitives is plotted.

**Four Room Maze:** We consider the Four-rooms gridworld environment (Sutton et al., 1999c) where the agent has to navigate its way through a grid of four interconnected rooms to reach a goal position within the grid. We consider the scenario where the starting position of the agent is fixed, but the goal is sampled from a discrete set. Fig. 3 shows that the proposed algorithm can learn four primitives. Refer to Appendix F for more details.

**Motion Imitation.** To evaluate the proposed method in terms of scalability, we present a series of tasks from the motion imitation domain, showing that we can use a set of distinct primitives for imitation learning. In these tasks, we train a simulated 2D biped character to perform a variety of highly dynamic skills by imitating motion capture clips recorded from human actors. Each mocap

https://github.com/jeanharb/option_critic
https://github.com/openai/mlsh

clip is represented by a target state trajectory $\tau^* = \{s_0^*, s_1^*, ..., s_T^*\}$, where $s_t^*$ denotes the target state at timestep $t$. The input to the policy is augmented with a goal $g_t = \{s_{t+1}^*, s_{t+2}^*\}$, which specifies the the target states for the next two timesteps. Both the state $s_t$ and goal $g_t$ are then processed by the encoder $p_{\text{enc}}(z_t|s_t, g_t)$. The repertoire of skills consists of 8 clips depicting different types of walks, runs, jumps, and flips. The motion imitation approach closely follows Peng et al. (2018). To analyze the specialization of the various primitives, we computed 2D embeddings of states and goals which each primitive is active in, and the actions proposed by the primitives. Fig. 4 illustrates the embeddings computed with t-SNE (van der Maaten & Hinton, 2008). The embeddings show distinct clusters for the primitives, suggesting a degree of specialization of each primitive to certain states, goals, and actions.

## 5.2 Multi-Task Training

We evaluate our model in a partially-observable 2D multi-task environment called Minigrid, similar to the one introduced in (Chevalier-Boisvert et al., 2018). The environment is a two-dimensional grid with a single agent, impassable walls, and many objects scattered in the environment. The agent is provided with a natural language string that specifies the task that the agent needs to complete. The setup is partially observable, and the agent only gets the small, egocentric view of the grid (along with the natural language task description). We consider three tasks here: the *Pickup* task (A), where the agent is required to pick up an object specified by the goal string, the *Unlock* task (B) where the agent needs to unlock the door (there could be multiple keys in the environment, and the agent needs to use the key which matches the color of the door) and the *UnlockPickup* task (C), where the agent first needs to unlock a door that leads to another room. In this room, the agent needs to find and pick up the object specified by the goal string. Additional implementation details of the environment are provided in appendix D. Details on the agent model can be found in appendix D.3.

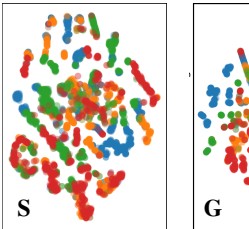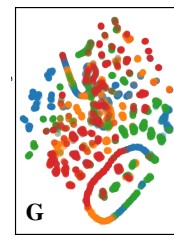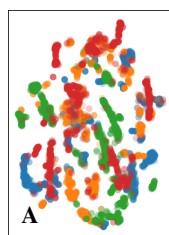

**S**    **G**    **A**

Figure 4: Embeddings visualizing the states (S) and goals (G) which each primitive is active in, and the actions (A) proposed by the primitives for the motion imitation tasks. A total of four primitives are trained. The primitives produce distinct clusters.

We train agents with varying numbers of primitives on various tasks – concurrently, as well as in transfer settings. The different experiments are summarized in Figs. 5 and 7. An advantage of the multi-task setting is that it allows for quantitative interpretability as to when and which primitives are being used. The results indicate that a system composed of multiple primitives generalizes more easily to a new task, as compared to a single policy. We further demonstrate that several primitives can be combined dynamically and that the individual primitives respond to stimuli from new environments when trained on related environments.

## 5.3 Do Learned Primitives Help in Transfer Learning?

We evaluate our approach in the settings where the adaptation to the changes in the task is vital. The argument in favor of modularity is that it enables better knowledge transfer between related tasks. Naturally, the transfer is easier when the tasks are closely related, as the model will only need to learn how to compose the already-learned primitives. In general, it is difficult to determine how closely related two tasks are, however, and the inductive bias of modularity could even be harmful if the two tasks are very different. In such cases, we could add new primitives (which would need to be learned) and still obtain a sample-efficient transfer, as some part of the task structure would already have been captured by the pretrained primitives. This approach can be extended towards adding primitives during training, providing a seamless way to combine knowledge about different tasks to solve more complex tasks. We investigate here the transfer properties of a primitive trained in one environment and transferred to a different one. Results are shown in Fig. 5.

**Continuous control for ant maze**    We evaluate the transfer performance of pretrained primitives on the cross maze environment (Haarnoja et al., 2018). Here, a quadrupedal ant robot must walk to the different goals along the different paths (see Appendix G for details). The goal is randomly

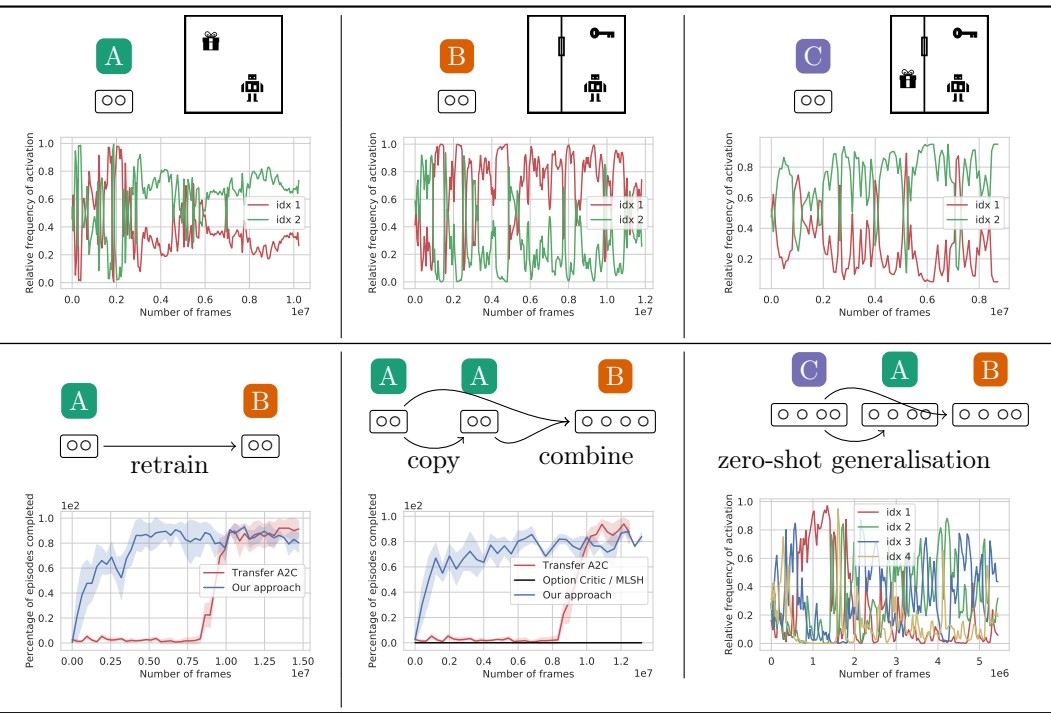

Figure 5: **Multitask training**. Each panel corresponds to a different training setup, where different tasks are denoted A, B, C, ..., and a rectangle with $n$ circles corresponds to an agent composed of $n$ primitives trained on the respective tasks. Top row: activation of primitives for agents trained on single tasks. Bottom row: **Retrain:** Two primitives are trained on task A and transferred to task B. The results (success rates) indicate that the multi-primitive model is substantially more sample efficient than the baseline (transfer A2C). **Copy and Combine:** More primitives are added to the model over time in a plug-and-play fashion (two primitives are trained on task A; the model is extended with a copy of the two primitives; the resulting four-primitive model is trained on task B.) This is more sample efficient than other strong baselines, such as (Frans et al., 2017; Bacon et al., 2017). **Zero-Shot Generalization:** A set of primitives is trained on task C, and zero-shot generalization to task A and B is evaluated. The primitives learn a form of spatial decomposition which allows them to be active in both target tasks, A and B. The checkpointed model is ran on 100 different episodes, and the normalized frequency of activation of the different primitives is plotted.

chosen from a set of available goals at the start of each episode. We pretrain a policy (see model details in Appendix G.1) with a motion reward in an environment which does not have any walls (similar to Haarnoja et al. (2018)), and then transfer the policy to the second task where the ant has to navigate to a random goal chosen from one of the 3 (or 10) available goal options. For our model, we make four copies of the pretrained policies and then finetune the model using the pretrained policies as primitives. We compare to both MLSH (Frans et al., 2017) and option-critic (Bacon et al., 2017). All these baselines have been pretrained in the same manner. As evident from Fig. 7, our method outperforms the other approaches. The fact that the initial policies successfully adapt to the transfer environment underlines the flexibility of our approach.

**Zero Shot Generalization:** The purpose of this experiment is to show that the model consisting of multiple primitives is somewhat able to decompose the task C into its subtasks, A and B. The better this decomposition is the better should the model transfer to the individual subtasks. In order to test this, we trained a set of 4 primitives on task C, and then evaluate them (without finetuning) on tasks A and B. We note that the ensemble is able to solve the transfer tasks, A and B, successfully 72% of the time, while a monolithic policy's success rate is 38%. This further shows that the primitives learn meaningful decompositions.

**Continual Learning: 4 Rooms Scneario.** We consider a continual learning scenario where we train two primitives for two-goal positions ie the goal position is selected randomly from one of the two positions at the start of the episode. The primitives are then transfer (and finetuned) on four-goal

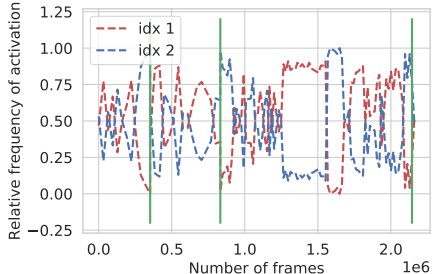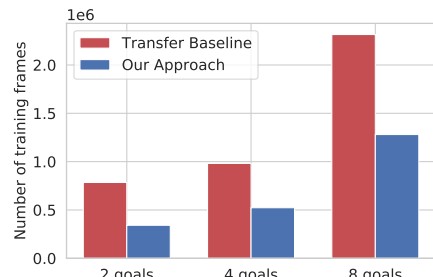

Figure 6: **Continual Learning Scenario:** The plot on the left shows that the primitives remain activated. The solid green line shows the boundary between the tasks. The plot on the right shows the number of samples required by our model and the transfer baseline model across different tasks. We observe that the proposed model takes fewer steps than the baseline (an A2C policy trained in a similar way), and the gap in terms of the number of samples keeps increasing as tasks become harder. The checkpointed model is ran on 100 different episodes (after a fixed number of steps/updates) and the normalized frequency of activation of the different primitives is plotted.

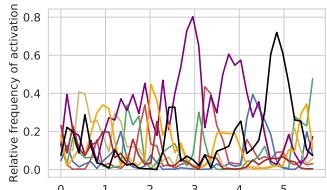

| Method | 3 goals | 10 goals |
|---|---|---|
| Flat Policy (PPO) | $11 \pm 5\ \%$ | $4 \pm 2\ \%$ |
| Option critic | $18 \pm 10\ \%$ | $7 \pm 3\ \%$ |
| MLSH | $32 \pm 3\ \%$ | $5 \pm 3\ \%$ |
| Explicit high level policy | $21 \pm 5\ \%$ | $11 \pm 2\ \%$ |
| **Proposed method** | $68 \pm 3\%$ | $40 \pm 3\%$ |

Figure 7: Left: Multitask setup, where we show that we are able to train eight primitives when training on a mixture of four tasks in the Minigrid environment. Here, the *x-axis* denotes the number of frames (timesteps). Right: Success rates of the different methods on the Ant Maze tasks. Success rate is measured as the number of times the ant is able to reach the goal (based on 500 sampled trajectories).

positions then transfer (and finetune) on eight-goal positions. The results are shown in fig. 6. The proposed method achieves better sample efficiency as compared to training a single monolithic policy.

# 6 SUMMARY AND DISCUSSION

We present a framework for learning an ensemble of primitive policies that can collectively solve tasks without learning an explicit master policy. Rather than relying on a centralized, learned meta-controller, the selection of active primitives is implemented through an information-theoretic mechanism. The learned primitives can be flexibly recombined to solve more complex tasks. Our experiments show that, on a partially observed "Minigrid" task and a continuous control "Ant Maze" walking task, our method can enable better transfer than flat policies and hierarchical RL baselines, including the Meta-learning Shared Hierarchies model and the Option-Critic framework. On Minigrid, we show how primitives trained with our method can transfer much more successfully to new tasks. On the Ant Maze, we show that primitives initialized from a pretrained walking control can learn to walk to different goals in a stochastic, multi-modal environment with nearly twice the success rate of a more conventional hierarchical RL approach, which uses the same pretraining but a centralized high-level policy. The proposed framework could be very attractive for continual learning settings, where one could add more primitive policies over time. Thereby, the already learned primitives would keep their focus on particular aspects of the task, and newly added ones could specialize on novel aspects.

## 7    ACKNOWLEDGEMENTS

The authors acknowledge the important role played by their colleagues at Mila throughout the duration of this work. AG would like to thank Greg Wayne, Mike Mozer, Matthew Botvinick, Bernhard Schölkopf for very useful discussions. The authors would also like to thank Nasim Rahaman, Samarth Sinha, Nithin Vasisth, Hugo Larochelle, Jordan Hoffman, Ankesh Anand, Michael Chang for feedback on the draft. The authors are grateful to NSERC, CIFAR, Google, Samsung, Nuance, IBM, Canada Research Chairs, Canada Graduate Scholarship Program, Nvidia for funding, and Compute Canada for computing resources. We are very grateful to Google for giving Google Cloud credits used in this project.

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

## A  INTERPRETATION OF THE REGULARIZATION TERM

The regularization term is given by

$$\mathcal{L}_{reg} = \sum_k \alpha_k \mathcal{L}_k \,,$$

where

$$\alpha_k = e^{\mathcal{L}_k} / \sum_j e^{\mathcal{L}_j} \,,$$

and thus

$$\log \alpha_k = \mathcal{L}_k - \log \sum_j e^{\mathcal{L}_j} \,,$$

or

$$\mathcal{L}_k = \log \alpha_k + \text{LSE}(\mathcal{L}_1, \dots, \mathcal{L}_K) \,,$$

where $\text{LSE}(\mathcal{L}_1, \dots, \mathcal{L}_K) = \log \sum_j e^{\mathcal{L}_j}$ is independent of $k$.

Plugging this in, and using $\sum \alpha_k = 1$, we get

$$\mathcal{L}_{reg} = \sum_k \alpha_k \log \alpha_k + \text{LSE}(\mathcal{L}_1, \dots, \mathcal{L}_K) = -H(\alpha) + \text{LSE}(\mathcal{L}_1, \dots, \mathcal{L}_K) \,.$$

**Information-theoretic interpretation**   Notably, $\mathcal{L}_{\text{reg}}$ also represents an upper bound to the KL-divergence of a mixture of the currently active primitives and a prior,

$$\mathcal{L}_{\text{reg}} \geq \text{D}_{\text{KL}}(\sum_k \alpha_k p_{\text{enc}}(Z_k|S) || \mathcal{N}(0,1)) \,,$$

and thus can be regarded as a term limiting the information content of the mixture of all active primitives. This arises from the convexity properties of the KL divergence, which directly lead to

$$\text{D}_{\text{KL}}(\sum_k \alpha_k f_k || g) \leq \sum_k \alpha_k \text{D}_{\text{KL}}(f_k || g) \,.$$

## B  ADDITIONAL RESULTS

### B.1  2D BANDITS ENVIRONMENT

In order to test if our approach can learn distinct primitives, we used the 2D moving bandits tasks (introduced in Frans et al. (2017)). In this task, the agent is placed in a 2D world and is shown the position of two randomly placed points. One of these points is the goal point but the agent does not know which. We use the sparse reward setup where the agent receives the reward of 1 if it is within a certain distance of the goal point and 0 at all other times. Each episode lasts for 50 steps and to get the reward, the learning agent must reach near the goal point in those 50 steps. The agent's action space consists of 5 actions - moving in one of the four cardinal directions (top, down, left, right) and staying still.

### B.1.1  RESULTS FOR 2D BANDITS

We want to answer the following questions:

1. Can our proposed approach learn primitives which remain active throughout training?
2. Can our proposed approach learn primitives which can solve the task?

We train two primitives on the 2D Bandits tasks and evaluate the relative frequency of activation of the primitives throughout the training. It is important that both the primitives remain active. If only 1 primitive is acting most of the time, its effect would be the same as training a flat policy. We evaluate the effectiveness of our model by comparing the success rate with a flat A2C baseline. Fig. 8 shows that not only do both the primitives remain active throughout training, our approach also outperforms the baseline approach.

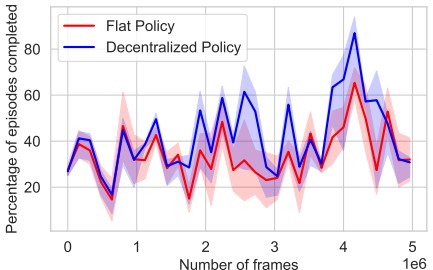 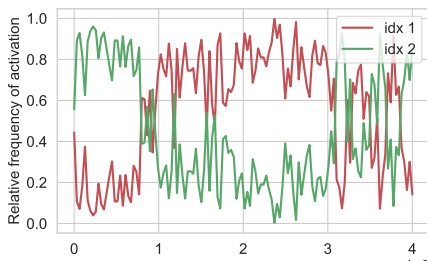

Figure 8: Performance on the 2D bandits task. Left: The comparison of our model (blue curve - decentralized policy) with the baseline (red curve - flat policy) in terms of success rate shows the effectiveness of our proposed approach. Right: Relative frequency of activation of the primitives (normalized to sum up to 1). Both primitives are utilized throughout the training.

## B.2 FOUR-ROOMS ENVIRONMENT

We consider the Four-rooms gridworld environment (Sutton et al., 1999c) where the agent has to navigate its way through a grid of four interconnected rooms to reach a goal position within the grid. The agent can perform one of the following four actions: *move up*, *move down*, *move left*, *move right*. The environment is stochastic and with 1/3 probability, the agent's chosen action is ignored and a new action (randomly selected from the remaining 3 actions) is executed ie the agent's selected action is executed with a probability of only 2/3 and the agent takes any of the 3 remaining actions with a probability of 1/9 each.

### B.2.1 TASK DISTRIBUTION FOR THE FOUR-ROOM ENVIRONMENT

In the Four-room environment, the agent has to navigate to a goal position which is randomly selected from a set of goal positions. We can use the size of this set of goal positions to define a curriculum of task distributions. Since the environment does not provide any information about the goal state, the larger the goal set, harder is the task as the now goal could be any element from a larger set. The choice of the set of goal states and the choice of curriculum does not affect the environment dynamics. Specifically, we consider three tasks - *Fourroom-v0*, *Fourroom-v1* and *Fourroom-v2* with the set of 2, 4 and 8 goal positions respectively. The set of goal positions for each task is fixed but not known to the learning agent. We expect, and empirically verify, that the *Fourroom-v0* environment requires the least number of samples to be learned, followed by the *Fourroom-v1* and the *Fourroom-v2* environment (Fig. 6 in the paper).

### B.2.2 RESULTS FOR FOUR-ROOMS ENVIRONMENT

We want to answer the following questions:

1. Can our proposed approach learn primitives that remain active when training the agent over a sequence of tasks?
2. Can our proposed approach be used to improve the sample efficiency of the agent over a sequence of tasks?

To answer these questions, we consider two setups. In the baseline setup, we train a flat A2C policy on *Fourrooms-v0* till it achieves a 100 % success rate during evaluation. Then we transfer this policy to *Fourrooms-v1* and continue to train till it achieves a 100 % success rate during the evaluation on *Fourrooms-v1*. We transfer the policy one more time to *Fourrooms-v2* and continue to train the policy until it reaches a 60% success rate. In the last task(*Fourrooms-v2*), we do not use 100% as the threshold as the models do not achieve 100% for training even after training for 10M frames. We use 60% as the baseline models generally converge around this value.

In the second setup, we repeat this exercise of training on one task and transferring to the next task with our proposed model. Note that even though our proposed model converges to a higher value than 60% in the last task(*Fourrooms-v2*), we compare the number of samples required to reach 60% success rate to provide a fair comparison with the baseline.

## C    IMPLEMENTATION DETAILS

In this section, we describe the implementation details which are common for all the models. Other task-specific details are covered in the respective task sections.

1. All the models (proposed as well as the baselines) are implemented in Pytorch 1.1 unless stated otherwise. (Paszke et al., 2017).

2. For Meta-Learning Shared Hierarchies (Frans et al., 2017) and Option-Critic (Bacon et al., 2017), we adapted the author's implementations for our environments.

3. During the evaluation, we use 10 processes in parallel to run 500 episodes and compute the percentage of times the agent solves the task within the prescribed time limit. This metric is referred to as the "success rate".

4. The default time limit is 500 steps for all the tasks unless specified otherwise.

5. All the feedforward networks are initialized with the *orthogonal* initialization where the input tensor is filled with a (semi) orthogonal matrix.

6. For all the embedding layers, the weights are initialized using the unit-Gaussian distribution.

7. The weights and biases for all the GRU model are initialized using the uniform distribution from $U(-\sqrt{k}, \sqrt{k})$ where $k = \frac{1}{hidden\_size}$.

8. During training, we perform 64 rollouts in parallel to collect 5-step trajectories.

9. The $\beta_{ind}$ and $\beta_{reg}$ parameters are both selected from the set $\{0.001, 0.005, 0.009\}$ by performing validation.

In section D.4.2, we explain all the components of the model architecture along with the implementation details in the context of the MiniGrid Environment. For the subsequent environments, we describe only those components and implementation details which are different than their counterpart in the MiniGrid setup and do not describe the components which work identically.

## D    MINIGRID ENVIRONMENT

We use the MiniGrid environment (Chevalier-Boisvert et al., 2018) which is an open-source, grid-world environment package . It provides a family of customizable reinforcement learning environments that are compatible with the OpenAI Gym framework (Brockman et al., 2016). Since the environments can be easily extended and modified, it is straightforward to control the complexity of the task (eg controlling the size of the grid, the number of rooms or the number of objects in the grid, etc). Such flexibility is very useful when experimenting with curriculum learning or testing for generalization.

### D.1    THE WORLD

In MiniGrid, the world (environment for the learning agent) is a rectangular grid of size say $MxN$. Each tile in the grid contains either zero or one object. The possible object types are *wall*, *floor*, *lava*, *door*, *key*, *ball*, *box* and *goal*. Each object has an associated string (which denote the object type) and an associated discrete color (could be red, green, blue, purple, yellow and grey). By default, walls are always grey and goal squares are always green. Certain objects have special effects. For example, a key can unlock a door of the same color.

### D.1.1    REWARD FUNCTION

We consider the sparse reward setup where the agent gets a reward (of 1) only if it completes the task and 0 at all other time steps. We also apply a time limit of 500 steps on all the tasks ie the agent must complete the task in 500 steps. A task is terminated either when the agent solves the task or when the time limit is reached - whichever happens first.

---

https://github.com/openai/mlsh, https://github.com/jeanharb/option_critic
https://github.com/maximecb/gym-minigrid

### D.1.2 ACTION SPACE

The agent can perform one of the following seven actions per timestep: *turn left*, *turn right*, *move forward*, *pick up an object*, *drop the object being carried*, *toggle*, *done* (optional action).

The agent can use the *turn left* and *turn right* actions to rotate around and face one of the 4 possible directions (north, south, east, west). The *move forward* action makes the agent move from its current tile onto the tile in the direction it is currently facing, provided there is nothing on that tile, or that the tile contains an open door. The *toggle* actions enable the agent to interact with other objects in the world. For example, the agent can use the *toggle* action to open the door if they are right in front of it and have the key of matching color.

### D.1.3 OBSERVATION SPACE

The MiniGrid environment provides partial and egocentric observations. For all our experiments, the agent sees the view of a square of 4x4 tiles in the direction it is facing. The view includes the tile on which the agent is standing. The observations are provided as a tensor of shape $4x4x3$. However, note that this tensor does not represent RGB images. The first two channels denote the view size and the third channel encodes three integer values. The first integer value describes the type of the object, the second value describes the color of the object and the third value describes if the doors are open or closed. The benefit of using this encoding over the RGB encoding is that this encoding is more space-efficient and enables faster training. For human viewing, the fully observable, RGB image view of the environments is also provided and we use that view as an example in the paper.

Additionally, the environment also provides a natural language description of the goal. An example of the goal description is: "Unlock the door and pick up the red ball". The learning agent and the environment use a shared vocabulary where different words are assigned numbers and the environment provides a number-encoded goal description along with each observation. Since different instructions can be of different lengths, the environment pads the goal description with *<unk>* tokens to ensure that the sequence length is the same. When encoding the instruction, the agent ignores the padded sub-sequence in the instruction.

### D.2 TASKS IN MINIGRID ENVIRONMENT

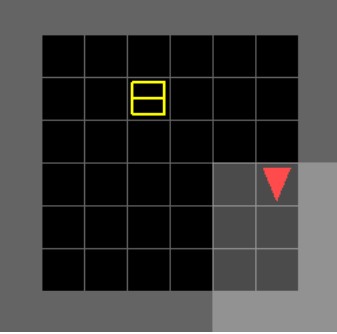

Figure 9: RGB view of the *Pickup* environment.

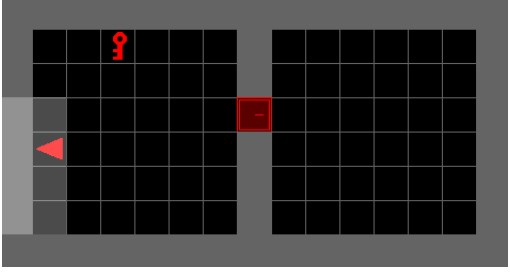

Figure 10: RGB view of the *Unlock* environment.

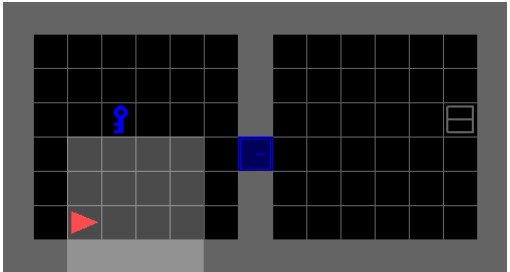

Figure 11: RGB view of the *UnlockPickup* environment.

We consider the following tasks in the MiniGrid environment:

1. **Pickup**: In the *Pickup* task, the agent spawns at an arbitrary position in a $8 \times 8$ grid (Fig. 9 ). It is provided with a natural language goal description of the form "go pickup a yellow box". The agent has to navigate to the object being referred to in the goal description and pick it up.

2. **Unlock**: In the *Unlock* task, the agent spawns at an arbitrary position in a two-room grid environment. Each room is $8 \times 8$ square (Fig. 10 ). It is provided with a natural language goal description of the form "open the door". The agent has to find the key that corresponds to the color of the door, navigate to that key and use that key to open the door.

3. **UnlockPickup**: This task is basically a union of the *Unlock* and the *Pickup* tasks. The agent spawns at an arbitrary position in a two-room grid environment. Each room is $8 \times 8$ square (Fig. 11 ). It is provided with a natural language goal description of the form "open the door and pick up the yellow box". The agent has to find the key that corresponds to the color of the door, navigate to that key, use that key to open the door, enter the other room and pick up the object mentioned in the goal description.

## D.3 MODEL ARCHITECTURE

### D.3.1 TRAINING SETUP

Consider an agent training on any task in the MiniGrid suite of environments. At the beginning of an episode, the learning agent spawns at a random position. At each step, the environment provides observations in two modalities - a $4 \times 4 \times 3$ tensor $x_t$ (an egocentric view of the state of the environment) and a variable length goal description $g$. We describe the design of the learning agent in terms of an encoder-decoder architecture.

### D.3.2 ENCODER ARCHITECTURE

The agent's *encoder* network consists of two models - a CNN+GRU based *observation encoder* and a GRU (Cho et al., 2014) based *goal encoder*

**Observation Encoder**:

It is a three layer CNN with the output channel sizes set to 16, 16 and 32 respectively (with ReLU layers in between) and kernel size set to $2 \times 2$ for all the layers. The output of the CNN is flattened and fed to a GRU model (referred to as the *observation-rnn*) with 128-dimensional hidden state. The output from the *observation-rnn* represents the encoding of the observation.

**Goal Encoder**:

It comprises of an embedding layer followed by a unidirectional GRU model. The dimension of the embedding layer and the hidden and the output layer of the GRU model are all set to 128.

The concatenated output of the *observation encoder* and the *goal encoder* represents the output of the *encoder*.

### D.3.3 DECODER

The decoder network comprises the *action network* and the *critic network* - both of which are implemented as feedforward networks. We now describe the design of these networks.

### D.3.4 VALUE NETWORK

1. Two-layer feedforward network with the tanh non-linearity.

2. Input: Concatenation of $z$ and the current hidden state of the *observation-rnn*.

3. Size of the input to the first layer and the second layer of the *policy network* are 320 and 64 respectively.

4. Produces a scalar output.

## D.4 COMPONENTS SPECIFIC TO THE PROPOSED MODEL

The components that we described so far are used by both the baselines as well as our proposed model. We now describe the components that are specific to our proposed model. Our proposed model consists of an ensemble of primitives and the components we describe apply to each of those primitives.

### D.4.1 INFORMATION BOTTLENECK

Given that we want to control and regularize the amount of information that the *encoder* encodes, we compute the KL divergence between the output of the *action-feature encoder network* and a diagonal unit Gaussian distribution. More is the KL divergence, more is the information that is being encoded with respect to the Gaussian prior and vice-versa. Thus we regularize the primitives to minimize the KL divergence.

### D.4.2 HYPERPARAMETERS

Table 1 lists the different hyperparameters for the MiniGrid tasks.

| Parameter | Value |
|---|---|
| Learning Algorithm | A2C (Wu et al., 2017) |
| Opitimizer ' | RMSProp (Tieleman & Hinton, 2012) |
| learning rate | $7 \cdot 10^{-4}$ |
| batch size | 64 |
| discount | 0.99 |
| lambda (for GAE (Schulman et al., 2015)) | 0.95 |
| entropy coefficient | $10^{-2}$ |
| loss coefficient | 0.5 |
| Maximum gradient norm | 0.5 |

Table 1: Hyperparameters

## E 2D BANDITS ENVIRONMENT

### E.0.1 OBSERVATION SPACE

The 2D bandits task provides a 6-dimensional flat observation. The first two dimensions correspond to the $(x, y)$ coordinates of the current position of the agent and the remaining four dimensions correspond to the $(x, y)$ coordinates of the two randomly chosen points.

### E.1 MODEL ARCHITECTURE

### E.1.1 TRAINING SETUP

Consider an agent training on the 2D bandits tasks. The learning agent spawns at a fixed position and is randomly assigned two points. At each step, the environmental observation provides the current position of the agent as well the position of the two points. We describe the design of the learning agent in terms of an encoder-decoder architecture.

### E.1.2 ENCODER ARCHITECTURE

The agent's *encoder* network consists of a GRU-based recurrent model (referred as the *observation-rnn*) with a hidden state size of 128. The 6-dimensional observation from the environment is the input to the GRU model. The output from the *observation-rnn* represents the encoding of the observation.

### E.2 HYPERPARAMETERS

The implementation details for the 2D Bandits environment are the same as that for MiniGrid environment and are described in detail in section D.4.2. In the table below, we list the values of the task-specific hyperparameters.

| Parameter | Value |
|---|---:|
| Learning Algorithm | PPO (Schulman et al., 2017) |
| epochs per update (PPO) | 10 |
| Optimizer ' | Adam (Kingma & Ba, 2014) |
| learning rate | $3 \cdot 10^{-5}$ |
| $\beta_1$ | 0.9 |
| $\beta_2$ | 0.999 |
| batch size | 64 |
| discount | 0.99 |
| entropy coefficient | 0 |
| loss coefficient | 1.0 |
| Maximum gradient norm | 0.5 |

Table 2: Hyperparameters

## F FOUR-ROOMS ENVIRONMENT

### F.1 THE WORLD

In the Four-rooms setup, the world (environment for the learning agent) is a square grid of say $11 \times 11$. The grid is divided into 4 rooms such that each room is connected with two other rooms via hallways. The layout of the rooms is shown in Fig. 12. The agent spawns at a random position and has to navigate to a goal position within 500 steps.

### F.1.1 REWARD FUNCTION

We consider the sparse reward setup where the agent gets a reward (of 1) only if it completes the task (and reaches the goal position) and 0 at all other time steps. We also apply a time limit of 300 steps on all the tasks ie the agent must complete the task in 300 steps. A task is terminate either when the agent solves the task or when the time limit is reached - whichever happens first.

### F.1.2 OBSERVATION SPACE

The environment is a $11 \times 11$ grid divided into 4 interconnected rooms. As such, the environment has a total of 104 states (or cells) that can be occupied. These states are mapped to integer identifiers. At any time $t$, the environment observation is a one-hot representation of the identifier corresponding to

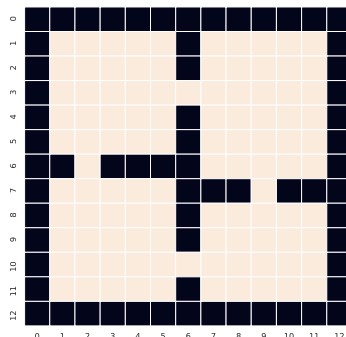

Figure 12: View of the four-room environment

the state (or the cell) the agent is in right now. ie the environment returns a vectors of zeros with only one entry being 1 and the index of this entry gives the current position of the agent. The environment does not return any information about the goal state.

## F.2 MODEL ARCHITECTURE FOR FOUR-ROOM ENVIRONMENT

### F.2.1 TRAINING SETUP

Consider an agent training on any task in the Four-room suite of environments. At the beginning of an episode, the learning agent spawns at a random position and the environment selects a goal position for the agent. At each step, the environment provides a one-hot representation of the agent's current position (without including any information about the goal state). We describe the design of the learning agent in terms of an encoder-decoder architecture.

## F.3 ENCODER ARCHITECTURE

The agent's *encoder* network consists of a GRU-based recurrent model (referred as the *observation-rnn* with a hidden state size of 128. The 104-dimensional one-hot input from the environment is fed to the GRU model. The output from the *observation-rnn* represents the encoding of the observation.

The implementation details for the Four-rooms environment are the same as that for MiniGrid environment and are described in detail in section D.4.2.

## G  ANT MAZE ENVIRONMENT

We use the Mujoco-based quadruple ant (Todorov et al., 2012) to evaluate the transfer performance of our approach on the cross maze environment (Haarnoja et al., 2018). The training happens in two phases. In the first phase, we train the ant to walk on a surface using a motion reward and using just 1 primitive. In the second phase, we make 4 copies of this trained policy and train the agent to navigate to a goal position in a maze (Figure 13). The goal position is chosen from a set of 3 (or 10) goals. The environment is a continuous control environment and the agent can directly manipulate the movement of joints and limbs.

### G.0.1 OBSERVATION SPACE

In the first phase (training the ant to walk), the observations from the environment correspond to the state-space representation ie a real-valued vector that describes the state of the ant in mechanical terms - position, velocity, acceleration, angle, etc of the joints and limbs. In the second phase (training the ant to navigate the maze), the observation from the environment also contains the location of the goal position along with the mechanical state of the ant.

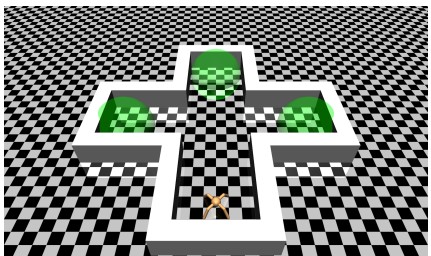

Figure 13: View of the Ant Maze environment with 3 goals

### G.1 MODEL ARCHITECTURE FOR ANT MAZE ENVIRONMENT

#### G.1.1 TRAINING SETUP

We describe the design of the learning agent in terms of an encoder-decoder architecture.

#### G.1.2 ENCODER ARCHITECTURE

The agent's *encoder* network consists of a GRU-based recurrent model (referred as the *observation-rnn* with a hidden state size of 128. The real-valued state vector from the environment is fed to the GRU model. The output from the *observation-rnn* represents the encoding of the observation. Note that in the case of phase 1 vs phase 2, only the size of the input to the *observation-rnn* changes and the encoder architecture remains the same.

#### G.1.3 DECODER

The decoder network comprises the *action network* and the *critic network*. All these networks are implemented as feedforward networks. The design of these networks is very similar to that of the *decoder* model for the MiniGrid environment as described in section D.3.3 with just one difference. In this case, the action space is continuous so the *action-feature decoder network* produces the mean and log-standard-deviation for a diagonal Gaussian policy. This is used to sample a real-valued action to execute in the environment.

