# OpenReview forum: "Reinforcement Learning with Competitive  Ensembles of Information-Constrained Primitives"
_ICLR.cc/2020/Conference — Accept (Poster)_

### Official Review · AnonReviewer3 · 2019-10-22
**Official Blind Review #3**

**Rating:** 6

**Review:**

This paper is about a policy design, where the policy is expressed as a mixture of policies called primitives.  Each primitive is made of an encoder and a decoder, mapping state to actions, rather than temporally extended actions (or options in RL).  The primitives compete with each other to be selected in each state and thus do away with the need for a meta-policy to select the primitives.  The selected primitive in each state trades between reward maximization and information content.

The paper is well written and is enjoyable to read.  It is helpful for me to have equation (3) in mind before reading about the explanation on the tradeoff between the reward and information, but this is a minor point.  My concern is that by scaling the reward in proportion to L_k redistributes the rewards in a way that is not reflective of the underlying reward structure of the MDP.  If so, the constructed policy \pi could place a high probability on the suboptimal actions.  How do we know if the action selected according to policy \pi will indeed lead to high rewards?

**Experience Assessment:**

I have read many papers in this area.

**Review Assessment: Checking Correctness Of Derivations And Theory:**

I assessed the sensibility of the derivations and theory.

**Review Assessment: Checking Correctness Of Experiments:**

I assessed the sensibility of the experiments.

**Review Assessment: Thoroughness In Paper Reading:**

I read the paper at least twice and used my best judgement in assessing the paper.

---

> ### Author Response · Authors · 2019-11-08
> **Order of presentation and reward scaling**
>
> We thank the reviewer for their feedback and their generally positive assessment of our work.
>
>
> “Equation 3”
>
> We thank the reviewer for pointing this out. We will update the order of the presentation.
>
>
> “Scaling the reward in proportion to L_k”
>
> Thanks for raising this point. We would like to point out that for the overall performance it is not necessarily important for individual primitives to be able to solve the task; what matters is whether the whole ensemble solves the task, which is what we show. As long as the whole ensemble is trained jointly on the overall reward, and the reward is substantially larger than any other regularizing losses, everything should be fine (as also confirmed by our empirical evaluation).

---

### Official Review · AnonReviewer2 · 2019-10-23
**Official Blind Review #2**

**Rating:** 6

**Review:**

The paper draws upon the idea of information bottleneck to do task decomposition so as to learn policy primitives similar to hierarchical reinforcement learning that combine together in a competitive manner to specialize in different parts of the task's domain. These policy primitives don't need a higher-level meta-policy to stitch them together. Instead the decision is made in a decentralized manner balancing the cost of information acquisition with maximizing rewards.
The paper seems to build on the idea of decomposing the task primitives that specialize in different parts of the state space. A related recent paper [1] which used similar ideas, albeit with a central coordinator to do task composition is missing from related works.

There are other issues with the paper as well.
- Decision making is not exactly "decentralized"? Computing $Z$ in $\alpha_k$, still requires the values from other primitives?
- Sec 5.1 seems tacked on. Motion Imitation is not RL. Experiment details about motion imitation unclear from description.
- In Figure 5: Zero-shot generalization, it's unclear from the plot whether it actually generalized to solve the tasks A and B. Relative frequency of activation are meaningless without reporting the actual performance on these environments. Where do the 4 indices come from? From the plot above there were only 2 indices but plot below has 4.
- Continuous control tasks required pretraining.

Overall the experiments seem a little underwhelming. More details on transfer performance without pretraining would be quite helpful. I like the idea of a competitive ensemble figuring out a useful task decomposition and using an information bottleneck like approach makes sense.

[1] https://dl.acm.org/citation.cfm?id=3331671

*Edit*: Updated the score after the reading the revision and the authors' responses.

**Experience Assessment:**

I have published one or two papers in this area.

**Review Assessment: Checking Correctness Of Derivations And Theory:**

I assessed the sensibility of the derivations and theory.

**Review Assessment: Checking Correctness Of Experiments:**

I carefully checked the experiments.

**Review Assessment: Thoroughness In Paper Reading:**

I read the paper at least twice and used my best judgement in assessing the paper.

---

> ### Author Response · Authors · 2019-11-08
> **Clarifications regarding the method and experiments**
>
>
> We thank the reviewer for their feedback and their generally positive assessment of our work.
>
>
> “A related recent paper”
>
> Thanks for pointing out the missing reference! We will update the related work section to include this.
>
>
> “Decision making is not exactly "decentralized"? Computing in Z, in a_k still requires the values from other primitives?”
>
> Thanks for pointing out this ambiguity. We mean decentralized in the sense that individual primitives can be trained and function individually, without a central meta controller. For instance, we can train a set of primitives on task distribution A, train a different set of primitives on task distribution B, and then  combine them together to solve task distribution C (as in fig. 5 of the paper, for instance). We realize that this terminology was unclear and will revise the paper accordingly. We will replace the term “decentralized” with “implicitly decentralized” or “implicit master policy”.
>
>
> “Motion Imitation is not RL”
>
> We agree with the reviewer. Imitation learning is a simpler problem as compared to reinforcement learning. In the given task, we intend to demonstrate that we can use a set of distinct primitives for imitation learning. We use the exact same experimental protocol as in [1]. We will expand the experiment description to clarify this.
>
> [1] Deepmimic: Example guided deep reinforcement learning of physics-based character skills (Peng et al.)
>
>
> “Zero-shot generalization”
>
> Thanks for pointing out the discrepancy in the figure. We realize that there was a mistake in the illustration in the “zero-shot generalization” panel of fig. 5: each model should consist of 4 primitives, rather than 2; the 4 traces in the plot are correct, however. The purpose of this experiment is to show that the model consisting of multiple primitives is somewhat able to decompose the task C into its subtasks, A and B. The better this decomposition the better should the model transfer to the individual subtasks. The performance obtained on this task, which we will include in the updated paper, is as follows:
>
> Train on task C                                                Generalization performance on tasks A and B (avg.)
>
> 4 policies (Proposed method)                      72%
> Monolithic Policy                                            38%
>
>
> “Continuous control tasks required pretraining.”
>
> The goal of the continuous control experiments is to show that we can repurpose the learnt primitives: we first pretrain the primitives to walk (i.e in the absence of any goal), and then transfer the trained primitives to a more complex environment (where they have to navigate to a set of distinct goals), where they are trained further. This task is challenging, as the policy gets a reward only on reaching the goals. We also compared the proposed method to other hierarchical baselines like MLSH, Option Critic, and Feudal-like architectures. While pretraining is not strictly required to solve this task, we are specifically interested in whether and how we can reuse already pretrained primitives.

---

> > ### Author Response · Authors · 2019-11-13
> > **Updated Manuscript**
> >
> > We have update the manuscript.  Please refer to the general post for details:
> > https://openreview.net/forum?id=ryxgJTEYDr&noteId=HJl6plZKoB
> >
> > We believe have addressed your concerns. Do you have an updated impression of our paper? Thanks for your consideration.

---

> > ### Comment · AnonReviewer2 · 2019-11-14
> > **Re: While pretraining is not strictly required**
> >
> > > While pretraining is not strictly required to solve this task
> >
> > Without experiments it's hard to say whether that's a true statement. Hierarchical methods have this chicken and egg problem of learning the basic policy vs specialization into different modules.

---

### Official Review · AnonReviewer1 · 2019-10-24
**Official Blind Review #1**

**Rating:** 8

**Review:**

	This paper takes a different approach for tackling the hierarchical RL problem. Their approach is to decompose the policy into a bunch of primitives. Each primitive acts according to its own interpretation of the state. All the primitives are competing with each other on a given state to take an action. It turns out that these primitive policies can be transferred to other tasks as they represent subtasks of a bigger task. The paper performs extensive experiments to show that this scheme improves over both flat and hierarchical policies in terms of generalization.

	This paper is well-written. I enjoyed reading it. Almost all my questions and doubts are explained when I read through the paper. The framework about using primitive policy to solve a big task is novel and original. The experiments are nice and convincing.

	Minor comments:
	• I am understanding the methods as a decomposing the policy into components. Different components are combined together using a probability distribution. To balance the competition and overlap of different primitives, different regularization objective are used.  Did you think about other simple methods, e.g., decompose the policy using linear combination work? It may worthwhile to compare your method with this baseline?
	• About the prior p(z), how to choose it in a meaningful way? I do not see why a unit gaussian is a good prior. It seems you want p(z|s) to be as close to a unit Gaussian as possible?
	• Figure 4 does not show a clear cluster structure. Explain more?


**Experience Assessment:**

I have published in this field for several years.

**Review Assessment: Checking Correctness Of Derivations And Theory:**

I assessed the sensibility of the derivations and theory.

**Review Assessment: Checking Correctness Of Experiments:**

I assessed the sensibility of the experiments.

**Review Assessment: Thoroughness In Paper Reading:**

I read the paper thoroughly.

---

> ### Author Response · Authors · 2019-11-08
> **Linear combination, Gaussian prior, and figure 4**
>
>
> We thank the reviewer for their feedback and their generally positive assessment of our work.
>
> Re linear combination: This is an interesting point. While in principle a policy could be defined as a linear combination of sub-policies, we are interested in the sparse case, where only one primitive becomes active at a time, and thus is the sole actor at that time. A linear combination of primitives would mean that many or all of them would be acting, which, in turn, would mean that all primitives are active to some extent at any time and therefore need to know how to act in any state. This is contradictory to our main objective of achieving specialized primitives that focus on certain parts of the state space.
>
> Re prior: While this choice is somewhat arbitrary and other distributions could be employed, Gaussian priors have been used extensively in latent variable models and in settings similar to ours (e.g. [1-3]) to introduce a certain geometry in latent space. The benefits of using a Gaussian prior are, among others, efficient evaluation and analytical tractability (e.g. the reparameterization trick and analytical computation of the KL term). In short, using a Gaussian prior is mainly a matter of convenience, but it is sufficient, as the nonlinear decoder can transform it into an arbitrary distribution.
>
> [1] Variational Information Bottleneck, https://arxiv.org/abs/1612.00410
> [2] InfoBot, https://arxiv.org/abs/1901.10902
> [3] Information Asymmetry in KL regularized RL, https://arxiv.org/abs/1905.01240
>
>
> Re figure 4: We agree that the visualization does not show a very clear structure. This is likely because there is no natural way of clustering the states of the given task. Nevertheless, many of the small clusters that emerge are mostly taken care of by a single primitive (single color in the diagram.)
> Fig. 2 in the main paper is perhaps a more intuitive way of visualizing the dynamics: in the bottom panel it can be seen that individual primitives are active for certain sequences of consecutive states.
> In addition to the static material, we provide a video showing the evolution of the dynamics. The 4 blue bars indicate the relative activation of the different primitives in the current state.
>
> Video: https://gofile.io/?c=8hZkaW

---

> > ### Author Response · Authors · 2019-11-13
> > **Updated Manuscript.**
> >
> > We have update the manuscript.  Please refer to the general post for details:
> > https://openreview.net/forum?id=ryxgJTEYDr&noteId=HJl6plZKoB
> >
> > We believe have addressed your concerns and clarified some of your points. Do you have an updated impression of our paper? Thanks for your consideration and time. Appreciate it.

---

### Author Response · Authors · 2019-11-13
**Updated manuscript**


We have uploaded an updated version of the manuscript, taking into account the suggestions made by the reviewers.

In particular, we

- added Wu et al. (2019) to the related work section;

- corrected the bottom right panel of figure 5;

- now use a more nuanced notion of "decentralized", for instance, replacing
  - "fully decentralized approach" -> "decentralized approach",
  - "without learning a high-level controller" -> "without learning an explicit high-level controller",
  - and adding a clarifying statement to the introduction: "While, technically, the competition between primitives implicitly realizes a global selection mechanism, we consider our model "decentralized" in the sense that individual primitives can function on their own, and can be combined in new ways, without relying on an explicit high-level controller."

- clarify the zero-shot experiment and add numerical results: "The purpose of this experiment is to show that the model consisting of multiple primitives is somewhat able to decompose the task C into its subtasks, A and B. The better this decomposition the better should the model transfer to the individual subtasks. In order to test this, we trained a set of 4 primitives on task C, and then evaluate them (without retraining) on tasks A and B. We note that the ensemble is able to solve the transfer tasks (A and B) successfully 72% of the time, while a monolithic policy's success rate is 38%."

- clarify the imitation learning experiment motivation: "[...] showing that we can use a set of distinct primitives for imitation learning."


We thank the reviewers again for their constructive feedback, and kindly ask them to update their evaluation, taking into account the changes made.

---

### Decision · Program_Chairs · 2019-12-19

**Decision:**

Accept (Poster)

**Comment:**

In contrast to many current hierarchical reinforcement learning approaches, the authors present a decentralized method that learns low level policies that decide for themselves whether to act in the current state, rather than having a centralized higher level meta policy that chooses between low level policies.  The reviewers primarily had minor concerns about clarity, reward scaling, and several other issues that were clarified by the authors.  The only outstanding concern is that of whether transfer/pretraining is required for the experiments to work or not.  While this is an interesting question that I would encourage authors to address as much as possible, it does not seem like a dealbreaker in light of the reviewers' agreement on the core contribution.  Thus, I recommend this paper for acceptance.